# Characteristics of perceived effective telesupervision practices: A case study of supervisees and supervisors

Priya Martin[1,2]*, Lucylynn Lizarondo[3], Saravana Kumar[4], Esther Jie Tian[4], Srinivas Kondalsamy-Chennakesavan[1], Geoff Argus[5,6]

1 Faculty of Medicine, Rural Clinical School, The University of Queensland, Toowoomba, Queensland, Australia, 2 Darling Downs Health, Baillie Henderson Hospital, Toowoomba, Queensland, Australia, 3 JBI, University of Adelaide, Adelaide, South Australia, Australia, 4 Allied Health and Human Performance, University of South Australia, Adelaide, South Australia, Australia, 5 Southern Queensland Rural Health, The University of Queensland, Toowoomba, Queensland, Australia, 6 School of Psychology and Wellbeing, University of Southern Queensland, Toowoomba, Queensland, Australia

* Priya.Martin@uq.edu.au

**Data Availability Statement:** The study is protected by ethics approval obtained from the Darling Downs Human Research Ethics Committee

## Abstract

### Introduction

Many healthcare workers have switched from face-to-face clinical supervision to telesupervision since the onset of the COVID-19 pandemic. Given the rise in prevalence of telesupervision and continuing remote working arrangements, telesupervision is no longer only limited to rural areas. As this remains an under-investigated area, this study aimed to explore supervisor and supervisee first hand experiences of effective telesupervision.

### Methods

A case study approach combining in-depth interviews of supervisors and supervisees, and document analysis of supervision documentation was used. De-identified interview data were analysed through a reflective thematic analysis approach.

### Results

Three supervisor-supervisee pairs from occupational therapy and physiotherapy provided data. Data analysis resulted in the development of four themes: Benefits vs limitations and risks, not often a solo endeavour, importance of face-to-face contact, and characteristics of effective telesupervision.

### Discussion

Findings of this study have confirmed that telesupervision is suited to supervisees and supervisors with specific characteristics, who can navigate the risks and limitations of this mode of clinical supervision. Healthcare organisations can ensure availability of evidence-informed training on effective telesupervision practices, as well as investigate the role of blended supervision models to mitigate some risks of telesupervision. Further studies could

for multi-sites. Participant consent to share the de-idetified dataset on a public repository was not obtained at the time of the study. Therefore, any reasonable request to access this dataset needs to be directed to the Darling Downs Human Research Ethics Committee via email at DDHHS-RESEARCH@health.qld.gov.au.

**Funding:** Declaration of funding This manuscript is a part of Dr Priya Martin's postdoctoral fellowship which was funded through an Advance Queensland Industry Research Fellowship Grant by the Queensland Government Department of Tourism, Innovation and Sport; and co-funded by Darling Downs Health, Southern Queensland Rural Health, the University of Queensland Rural Clinical School, and the University of South Australia. Grant number - AQIRF115-2020-CV. The funder (i.e., Queensland Government Department of Tourism, Innovation and Sport) had no role in study design, data collection and analysis, decision to publish, or preparation of the manuscript.

**Competing interests:** The authors have declared that no competing interests exist.

investigate the effectiveness of utilising additional professional support strategies that complement telesupervision, including in nursing and medicine, and ineffective telesupervision practices.

## Introduction

Telesupervision usage has increased since the onset of the COVID-19 pandemic given the physical distancing and isolation requirements. For healthcare workers in healthcare settings, clinical supervision remains a critical mechanism for obtaining support for their role (i.e., skills, knowledge, and competencies), and for their own mental health and wellbeing at work (i.e., coping with work stressors, and managing burnout) [1,2]. For several professions, clinical supervision is mandated through professional associations, registration guidelines, and/or organisational policies and directives. When face-to-face clinical supervision is not possible, technology such as Microsoft Teams, Zoom, phone, and e-mail tend to be used. This is referred to as telesupervision or internet supervision where supervisors and supervisees meet asynchronously via e-mail or webchat, or synchronously via videoconferencing or phone [3]. Despite the lack of face-to-face or in-person contact, telesupervision still needs to cater to all supervision functions or aspects, namely formative, normative, and restorative, in line with Proctor's tripartite model of supervision [4]. Formative functions include the supervisee gaining skills and knowledge (e.g., conducting an assessment, making a splint). Normative functions include clinical governance and quality assurance (e.g., compliance with policies, guidelines, protocols). The restorative function is dedicated to supporting the supervisee's emotional wellbeing at work [4].

Evidence on what makes telesupervision high quality and effective for practicing healthcare workers remains limited. A pre-pandemic systematic review on this topic found a scarcity of studies and called for further studies to add evidence to this area [5]. Another systematic review of physicians' perceptions of providing videoconference supervision and support to junior doctors in rural areas, also identified a scarcity of studies [6]. Available evidence suggests that telesupervision can be of high quality and effective if set up well [5]. Confirming this, a pre-pandemic multiple-baseline single-case design study in psychotherapy has showed that telesupervision can obtain comparable outcomes as that of in-person supervision, provided the supervisor is effective (i.e., open, supportive, and a good communicator), and technology used is familiar [7].

While pre-pandemic telesupervision research remained predominant in the mental health professions (e.g., psychology, counselling), and in the pre-registration or pre-qualification (i.e., student) and training space, studies of telesupervision practices of registered or qualified healthcare workers, especially in non-mental health areas, remain scarce. A qualitative study of the effectiveness of videoconferencing to support orthopaedic trainees in Queensland (Australia) highlighted the crucial role of selecting the right technology to maximise benefits [8]. Another pilot qualitative study conducted in Victoria (Australia) in 2012 explored views of registrars, supervisors, and patients on the use of a video camera for remote supervision, identifying some highlights and pitfalls. This dataset was re-analysed and published in light of the COVID-19 pandemic [9]. Xavier and colleagues [10] evaluated videoconference delivery of clinical supervision and education of 20 psycho-oncology staff. In this study the supervision component delivered by telephone was rated by 80% of participants as extremely satisfying. These participants also received complementary monthly educational sessions via videoconference [10]. It is noteworthy that all these studies have been conducted in rural areas.

However, as Simmons and colleagues [9] point out, telesupervision no longer remains an issue only pertinent to rural areas given its widespread use triggered by the pandemic.

Some information has been made available on telesupervision practices since the COVID-19 pandemic onset. A rapid review of eight COVID-19 studies on the disruptions to clinical supervision practices of healthcare workers and students in healthcare settings documented a surge in the use of telesupervision. Findings indicated that those with prior positive supervisory relationships remained largely unaffected by the switch from face-to-face to telesupervision. Findings also showed that some healthcare workers reported no impact on the quality of supervision following the switch, and that younger participants perceived telesupervision to be less effective compared to their older counterparts [11]. A multi-methods study of 144 health service psychology students that had used both hybrid or blended (i.e., telesupervision and in-person supervision), and telesupervision for clinical training in the United States reported high rates of satisfaction and indicated that telesupervision is highly acceptable and beneficial in this population for training purposes [12]. Examples are also available of COVID-19-related telesupervision practices to support students on placement [13,14]. A survey of healthcare workers in Queensland (n = 250) showed that 16% switched from face-to-face supervision to telesupervision owing to COVID-19, of which 76% reported satisfaction with the switch (unpublished). Given this increasing use of telesupervision and a dearth of evidence in this area, including in-depth exploration, this current study utilised a case study approach to explore factors that influence effective telesupervision practices from a supervisor and supervisee perspective.

## Materials and methods

### Research design

An overarching constructionist theoretical paradigm was used to allow for co-construction of knowledge and meaning in the given context. This allowed the researchers to hear all participant views with equal weighting [15]. An instrumental-use multiple-case method [16] was used where particular cases are studied to gain a broader appreciation of an issue [17]. This case study method was chosen as it focuses on generating useful and actionable findings, as well as identification of factors that explain the difference between what works and what doesn't work, which can then be used to inform decision making and to support policy and practice [16,18]. Within this case study method, utilisation-focused sampling was used which involved selecting cases that are relevant to the issues and decisions of concern to an identifiable group of stakeholders and intended users. This sampling allows purposive sampling strategies but adds a requirement that cases selected need to have credibility, relevance, and utility for primary intended users [16]. To meet this criterion, researchers actively engaged primary intended users in design and methods decisions, especially sampling. Consultation was undertaken with two workforce development officers from the study population at the study protocol development stage. Their feedback was incorporated in the study design and methods decisions. Participants were recruited through promotion at the end of previous survey studies in this population, whereby they were asked to contact the first author should they be interested in participating in the case study [19, unpublished].Further, a snowball sampling strategy was used additionally to recruit participants. Cases were studied concurrently to obtain a snapshot of the factors that influence effective telesupervision practices.

### Setting and participants

The study was advertised in four regional and rural Queensland public health services through organisational newsletters and social media. Information about the study was also provided

along with other clinical supervision research surveys to prompt those interested in this study to contact the principal investigator. Eligible participants were allied health professionals, doctors, nurses, and midwives.

## Data collection

Using a multi-methods approach, data from included cases were collected through in-depth interviews and supervision documentation. All interviews were conducted in September 2021 by the principal investigator (PM) between via videoconference. Interviews were audio-recorded and transcribed by an external transcription service provider. Telesupervision content in the documents was examined to determine if all three components of the Proctor's model (normative, formative and restorative) [4] were considered in the sessions to determine how much weightage each component received. Supervisor-supervisee pairs were provided with the option of participating in interviews together or separately, to account for the relationship dynamics and scheduling logistics. Except for one paired interview, the remaining participants opted for individual interviews. Supervision documentation was also obtained from consenting participants for a period of three months prior to the study. The interview guide (S1 Appendix) was informed by the results of previous clinical supervision studies in this population [19, unpublished].

## Data analysis

A reflexive thematic analysis approach was used to analyse data, where researchers play an integral role in knowledge production. In this approach, researchers familiarise themselves with the data, and continually question and query assumptions during data interpretation and coding [20]. Two researchers (PM and LL; healthcare workers with a background in occupational therapy (PM) and physiotherapy (LL); both content experts in clinical supervision, with extensive experience in clinical supervision training and research, and qualitative research methods) analysed the data independently and collaborated through regular discussions to develop themes. The supervision document content was mapped against three components of the Proctor's model [4].

## Ethics

The ethics approval for this study was obtained from Darling Downs Health Human Research Ethics Committee for multisites (Ref: HREA/2020/QTDD/69958; Date: 10/11/2020). Subsequently, site-specific approvals were obtained from all the participating organisations. Written, informed consent was obtained from all participants.

## Results

Participants included three supervisor-supervisee pairs (n = 6) from three participating health services. De-identified supervision documents were available from two of the three included pairs (n = 4). Participants were from occupational therapy and physiotherapy. All three supervisors and one supervisee were in senior roles, whereas two supervisees were employed in junior roles of which one was a recent graduate with five months post-graduation experience at the time of study. Frequency of supervision sessions ranged from fortnightly to every four to six weeks. Duration of sessions ranged from 45 to 60 minutes. At the time of the study, telesupervision arrangements had been in place between five months and two years. Only one of the three pairs used an agenda to guide the sessions, whereas two pairs had a supervision contract in place. All supervisors and supervisees had met each other face-to-face prior to commencing

telesupervision. All three supervisors had visited the respective supervisee's work area that was in a different geographical location. While none of the six participants had undertaken any specific training related to telesupervision, only one participant reported some awareness of best practice guidelines in this area. All participants were trained in general clinical supervision. Two of the three supervisees were working in a clinical role, with the third supervisee working in a combined clinical and management role. Two supervisors were working in education and training roles, and the third in a clinical role. All three pairs reported using both Teams and phone for their telesupervision sessions, dependent on the availability of a confidential space and computer for the session.

Thematic analysis of data from interviews and supervision documentation resulted in the development of four themes namely: Benefits vs limitations and risks, not often a solo endeavour, importance of face-to-face contact, and characteristics of effective telesupervision.

## 1. Benefits vs limitations and risks

Telesupervision was said to be well-suited for information and resource sharing, discussing challenges in the workplace, and for working across professions. Data from supervision documents indicated that most telesupervision content was related to Proctor's normative and restorative aspects, with limited attention to the formative aspect. There was agreement that formative functions attended to in telesupervision often took the form of clinical care discussions on the phone and relied more on the supervisee's ability to clearly describe the account. This was limited due to the inherent challenges of undertaking telehealth at the bedside, which requires extensive planning, preparation, and timely access to equipment. A physiotherapy supervisor of a new graduate, said:

> . . .We've had in the background as the ability for (supervisee) to watch me on telehealth whilst I am treating a patient, but actually haven't done that yet. . .it's been exercise point of view, or assessments of the shoulders. It is a little bit difficult to demonstrate because you usually have to put your hands on the shoulder, and I can't do that to myself. (Supervisor 1)

One supervisor well-summarised the limitations of telesupervision:

> While I think it meets a purpose, I do feel limited in what you can do, not what you can discuss, but what you can do. (Supervisor 2)

Another risk noted was a tendency to spend a lot of time in discussions that may side track other agenda items or to deprioritise a telesupervision session altogether. One supervisor noted:

> When it (supervision) is via telehealth, sometimes it is a little bit easier to deprioritise it. . .-when it is face-to-face or somebody is actually at your door, it kind of forces you. . .to make that time available. It's probably a little bit easy to not do that when it is via telehealth. (Supervisee 3)

## 2. Not often a solo endeavour

Participants noted that supervisees had other professional support arrangements in place to supplement telesupervision. This was especially considered important for supervisees holding clinical responsibilities, thereby needing more attention to skills and knowledge (i.e., formative

aspect). Participants noted that some aspects of physiotherapy require hands-on teaching, manipulation of joints, and muscles. Due to limitations in the telesupervision technology at hand, participants reported reliance on other measures such as practicing with another physio onsite, both supervisor and supervisee having a 'model' at their end to demonstrate and practice, and to hold off until scheduled face-to-face sessions. One supervisor described:

*. . .She (supervisee) wouldn't be able to access a private space to have Teams, to have confidentiality, to have a patient there with us to do a joint session together because of the set up at the rural facility. . .she's tried to circumvent that through work shadowing. . .at the (bigger regional) hub hospital (with another identified senior physiotherapist). (Supervisor 2)*

One supervisee recalled receiving additional supports from others outside the telesupervision arrangement based on the supervisor's recommendations:

*This year. . .I've met with different people. . .I've met with a research person, and a speech therapist from somewhere else. (Supervisee 2)*

Participants also noted that a combination of technology was needed to make telesupervision effective as all of them reported to using Teams (i.e., videoconference), phone, and email.

## 3. Importance of face-to-face contact

All supervisor-supervisee pairs had met each other face-to-face prior to commencing telesupervision. This was either through departmental meetings or professional development opportunities. All supervisors reported having undertaken site visits of the supervisees' work sites based in another geographical location. This not only ensured that the supervisor had a realistic understanding of the supervisee's work context, it also enhanced the supervisee's trust in the advice provided by the supervisor given their familiarity of the context including the team culture and politics. Almost all participants described their ideal telesupervision as having initial, as well as some ongoing face-to-face sessions to supplement telesupervision:

*I would like to get some occasional face-to-face supervision. (Supervisee 1)*

*We've known each other as colleagues for a long time (prior to telesupervision). (Supervisor 3)*

## 4. Characteristics of effective telesupervision

Participants noted several characteristics that lead to the perceptions of their telesupervision arrangements being effective.

**4.1. Planned, yet flexible.** All participants noted the importance of being planned and having a structure to guide their telesupervision sessions. This involved deciding on who is responsible for making the phone call, what technology will be used in the subsequent session, having a backup technology troubleshooting plan, using a supervision contract and agenda, and being intentional about the session. One supervisor outlined:

*I think that's really important, no matter what method you used. Booking in, scheduling in, and time is important because access to technology isn't always possible in all locations. (Supervisor 3)*

The importance of setting up telesupervision well at the outset was also reiterated:

*In the first session we might not get much supervision done. We might just spend time figuring out what it is going to look like for us, what the supervisee has access to, do they know how to use it. . .so that they've got a room, equipment they need, safe space. So that when we get together it's the best scenario. . .devoting time to that to start with. (Supervisor 2)*

One supervisee stressed the importance of flexibility:

*I am generally initiating that (supervision session) either via email or phone, or text message. . .whatever we've kind of got available at the time. . .Knowing that she (supervisor) is flexible, that we can adjust that supervision model as required, has made that successful. (Supervisee 3)*

**4.2. Right supervisor and right supervisee.** In line, with best practice in face-to-face clinical supervision, a 'right' supervisor for a telesupervision arrangement was noted to have these qualities: open and honest, supportive, good at feedback provision, non-judgemental, approachable, and accessible between sessions. One supervisor noted that providing timely feedback and openness were enablers of effective telesupervision:

*. . .Openness, so the ability to talk about anything, good and bad, making that easier from both points of view. It's not just the supervisee saying, "I think I stuffed up here", it's also the supervisor saying "I've looked at your timetable. What are you doing?" It's that ability to bring up good and bad. (Supervisor 1)*

Similarly, a 'right' supervisee for a telesupervision arrangement was noted to have these qualities: turning up to meetings and dialling in on time, comfortable taking about difficult and trivialist things, being proactive, able to describe clinical care situations comprehensively, and responsive to supervisor's feedback. One supervisor appreciated the proactiveness of her supervisee in obtaining additional support: *She has contacted other people to have one-off mentoring sessions. (Supervisor 2)*

## Discussion

Healthcare workers across the globe switched from face-to-face to telesupervision, due to physical isolation and social distancing requirements triggered by the COVID-19 pandemic. Given the dearth of literature on this topic, this study explored in-depth supervisor and supervisee experiences of effective telesupervision practices, using two data sources, to understand the factors underpinning them. While all participants perceived their telesupervision arrangements as being effective and successful, it is noteworthy that not all factors that influence effective clinical supervision or telesupervision were accounted for [1,2,5]. Gaps were noted in the use of supervision contracts, agendas, lack of formal training in and awareness of best practice telesupervision guidelines, lack of attention to formative aspects, and inadequate record keeping [1,2,5]. This study has highlighted a need for training healthcare workers in effective telesupervision practices. This is an area that healthcare organisations can invest in to ensure staff are trained in best telesupervision practices, given the recent increase in remote working and telesupervision usage.

This study confirms that a key risk in telesupervision is a lack of attention to the formative aspect of supervision, building on previous findings of a study of rural healthcare workers [21]. Telesupervision appears to lend itself well to normative and restorative aspects, creating an issue for 'hands-on' areas of practice as identified by physiotherapists in this study. This is

also a concern for recent graduates and supervisees that need more input into building their skills and knowledge. In the current study, all supervisees had other professional support strategies in place to overcome the limitations of telesupervision related to the formative aspect. This is in line with previous findings from the study of 16 healthcare workers from allied health that showed existence of other professional support measures [21]. As the pandemic has made telesupervision practices more prevalent, this is no longer an issue that is only pertinent to rural areas [9]. Upskilling healthcare workers in effective supervision practices is needed, so that they are more aware of the risks of telesupervision and can devise strategies to mitigate them. Although, participants in this study did not report using blended supervision models (combination of in-person and telesupervision components), this could be an option well-worth investigating [22,23]. Further studies can investigate the effectiveness of utilising additional professional support strategies that complement telesupervision.

This study confirms the importance of face-to-face contact in enhancing the perceived effectiveness of telesupervision. All participants in the current study had met with each other face-to-face and noted this to have influenced their decision to commence telesupervision. A previous systematic review noted face-to-face contact as a key influence in enhancing the quality and effectiveness of telesupervision, as it helps with building a positive supervisory relationship [5]. A COVID-19 study of 79 post-graduate trainees in social work and other healthcare professions also found that those that had prior face-to-face contact before switching into telesupervision remained largely unaffected by the switch [24]. Healthcare workers in the current study, in line with the previous study [24] indicated a preference for face-to-face or in-person supervision. Where this is not possible, and telesupervision is the only option, it is recommended to preference videoconference methods over the telephone to access non-verbal cues that are crucial in building rapport within the supervisory relationship [5]. Supplementary and opportunistic face-to-face sessions should be factored in where possible.

This study has highlighted the importance of a 'right' supervisor and 'right' supervisee for effective telesupervision. Participants in this study noted that supervisors relied heavily on supervisees' accounts of clinical case presentations or issues in the workplace that were discussed in the sessions. Therefore, the ideal supervisee for telesupervision needs to be driven, and be able to co-lead the relationship. Telesupervision may not be suited to supervisees that may be passive learners, those not motivated to learn, and those needing a lot of assistance with their roles. Similarly, the supervisor in the telesupervision arrangement needs to be familiar with the supervisee's work context, be approachable, accessible, supportive and non-judgemental. This is in line with previous findings from an interview study of 15 participants on blended supervision models to support postgraduate rural medical training in Australia [23], and another interview study of remote supervision with 11 participants [25]. Supervisees in telesupervision can be encouraged to co-lead the supervisory relationship by being better prepared for sessions, self-evaluating, negotiating objectives of supervision, and applying their learning expertise [26–28]. It is noteworthy that characteristics of a 'right' supervisor and 'right' supervisee for telesupervision is similar to that of face-to-face supervision arrangements. Previous research has noted that while there are similarities, a higher dosage of these characteristics or traits are required in telesupervision to make up for the absence of in-person contact [5,28].

## Strengths and limitations

At a time where telesupervision practices have rapidly risen, this research provides important evidence on telesupervision at the coal face, as well as provides strategies for healthcare workers, and organisations to promote effective telesupervision practices. The study recruited both

supervisors and supervisees and used different data sources to enable triangulation to enhance trustworthiness of findings. Although recruitment was open to all healthcare workers, the study included participants only from two allied health disciplines, necessitating further studies with healthcare workers from other allied health disciplines, nursing, midwifery and medicine. Recruiting healthcare workers to participate in research during this time period was challenging given the workforce issues triggered by the COVID-19 pandemic, hence this study only included three supervisor-supervisee pairs. However, this challenge was mitigated by using in-depth exploration and two sources of data to provide a thorough view, and through reaching data sufficiency. Further research can also investigate ineffective telesupervision practices to further understand the gaps.

## Conclusion

This study explored effective telesupervision practices through supervisor and supervisee perspectives using an in-depth case study approach. Findings of this study have confirmed that telesupervision is suited to supervisees and supervisors with specific characteristics, who are able to navigate the risks and limitations of this mode of clinical supervision. Healthcare organisations have a role to play in ensuring availability of evidence-informed training on effective telesupervision practices, as well as investigating the role of blended supervision models to mitigate some risks of telesupervision. Further studies could investigate the effectiveness of utilising additional professional support strategies that complement telesupervision, including in nursing and medicine, and ineffective telesupervision practices.

## Supporting information

**S1 Appendix. Interview guide.**
(DOCX)

## Acknowledgments

We would like to acknowledge Dr Tiana Gurney and Ms Martelle Ford for their supportive involvement in the first author's postdoctoral fellowship which this study is a part of. We thank all the case study participants.

## Author Contributions

**Conceptualization:** Priya Martin, Lucylynn Lizarondo, Saravana Kumar, Esther Jie Tian, Srinivas Kondalsamy-Chennakesavan, Geoff Argus.

**Formal analysis:** Priya Martin, Lucylynn Lizarondo.

**Funding acquisition:** Priya Martin, Saravana Kumar, Srinivas Kondalsamy-Chennakesavan, Geoff Argus.

**Investigation:** Priya Martin.

**Methodology:** Priya Martin, Lucylynn Lizarondo, Saravana Kumar, Esther Jie Tian, Srinivas Kondalsamy-Chennakesavan, Geoff Argus.

**Project administration:** Priya Martin.

**Supervision:** Priya Martin.

**Writing – original draft:** Priya Martin.

**Writing – review & editing:** Priya Martin, Lucylynn Lizarondo, Saravana Kumar, Esther Jie Tian, Srinivas Kondalsamy-Chennakesavan, Geoff Argus.

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
