## [Decision Letter · Decision Letter 0]

15 May 2023

PONE-D-22-33318Characteristics of perceived effective telesupervision practices: A case study of supervisees and supervisorsPLOS ONE

Dear Dr. Martin,

Thank you for submitting your manuscript to PLOS ONE. After careful consideration, we feel that it has merit but does not fully meet PLOS ONE’s publication criteria as it currently stands. Therefore, we invite you to submit a revised version of the manuscript that addresses the points raised during the review process.

We look forward to receiving your revised manuscript.

Kind regards,

Federica Canzan

Academic Editor

PLOS ONE

“Declaration of funding

This manuscript is a part of Dr Priya Martin's postdoctoral fellowship which was funded through an Advance Queensland Industry Research Fellowship Grant by the Queensland Government Department of Tourism, Innovation and Sport; and co-funded by Darling Downs Health, Southern Queensland Rural Health, the University of Queensland Rural Clinical School, and the University of South Australia. Grant number - AQIRF115-2020-CV.”

Reviewers' comments:

Reviewer's Responses to Questions

**Comments to the Author**

1. Is the manuscript technically sound, and do the data support the conclusions?

Reviewer #1: Yes

Reviewer #2: Partly

2. Has the statistical analysis been performed appropriately and rigorously? 

Reviewer #1: N/A

Reviewer #2: N/A

3. Have the authors made all data underlying the findings in their manuscript fully available?

Reviewer #1: Yes

Reviewer #2: Yes

4. Is the manuscript presented in an intelligible fashion and written in standard English?

Reviewer #1: Yes

Reviewer #2: Yes

5. Review Comments to the Author

Reviewer #1: Thanks for undertaking this important work that is going to be increasingly relevant in the new normal of a post-COVID world. I agree with the authors' observation that this approach may no longer just be confined to rural areas.

While the paper is generally well written I think there are a couple of areas that I think need a little more explanation:

- what are the specifics of the supervisory arrangements? For example they may be legislative (eg supervision of medical interns), educational (eg Registrars in general practice or teaching hospitals) or just good practice. Describing the context would be helpful in understanding the generalizability of the findings

- Proctor's tripartite model may not be familiar to all readers so I would suggest a brief explanation of the terminology used

Also, I note there is no mention of programs like the Remote Vocational Training Stream (https://rvts.org.au/) which has been using remote telesupervision for over 20 years.

The findings seem appropriate and grounded in the data, although I note there were only three sets of interviewees, which seems low. A comment on why this number was chosen, and whether data saturation was achieved may be worth considering.

The paper is generally well written and clear, although I wonder about the use of the term 'content experts' in line 111 - I think the expertise described is more process-based / educational than in the content being taught. Finally, line 253 says 'Several healthcare workers...' - I suspect it was more than this, and these words should be revised.

Thank you for the opportunity to review this paper.

Reviewer #2: Thank you for conducting this study. The topic is very interesting, current and useful for better understanding and utilizing telesupervision. Below are some suggestions:

-In the Introduction paragraph, I suggest providing more justification for the choice of Case study method and the Proctor model for data analysis

-In the Materials and Methods section, I suggest providing more explanation for the choice of the overarching constructionist theoretical paradigm and the sample selection method. It is not clear how the selecting cases was conducted. Was the snowball sampling strategy used? At the end, were the 6 participants the most relevant to the issues studied?

-Lastly, in the Results section, under the category 'Right supervisor and right supervisee,' I suggest highlighting the characteristics of the supervisor that are useful in telesupervision, as the characteristics listed (such as open and honest, supportive, good at feedback provision, non-judgmental, approachable) seem general, regardless of whether the supervision is remote or face-to-face.

6. PLOS authors have the option to publish the peer review history of their article (what does this mean?). If published, this will include your full peer review and any attached files.

Reviewer #1: **Yes: **Tarun Sen Gupta

Reviewer #2: No

---

## [Author Response · Author response to Decision Letter 0]

25 May 2023

Please see attached response letter.

---

## [Decision Letter · Decision Letter 1]

26 Jun 2023

Characteristics of perceived effective telesupervision practices: A case study of supervisees and supervisors

PONE-D-22-33318R1

Dear Dr. Martin,

We’re pleased to inform you that your manuscript has been judged scientifically suitable for publication and will be formally accepted for publication once it meets all outstanding technical requirements.

Kind regards,

Federica Canzan

Academic Editor

PLOS ONE

Additional Editor Comments (optional):

Reviewers' comments:

Reviewer's Responses to Questions

**Comments to the Author**

1. If the authors have adequately addressed your comments raised in a previous round of review and you feel that this manuscript is now acceptable for publication, you may indicate that here to bypass the “Comments to the Author” section, enter your conflict of interest statement in the “Confidential to Editor” section, and submit your "Accept" recommendation.

Reviewer #1: (No Response)

Reviewer #2: All comments have been addressed

2. Is the manuscript technically sound, and do the data support the conclusions?

Reviewer #1: Yes

Reviewer #2: Yes

3. Has the statistical analysis been performed appropriately and rigorously? 

Reviewer #1: Yes

Reviewer #2: N/A

4. Have the authors made all data underlying the findings in their manuscript fully available?

Reviewer #1: Yes

Reviewer #2: Yes

5. Is the manuscript presented in an intelligible fashion and written in standard English?

Reviewer #1: Yes

Reviewer #2: Yes

6. Review Comments to the Author

Reviewer #1: Thank you for addressing the reviewers' comments, I feel the manuscript is now stronger and an important addition to the literature. I note the response to my comment about citing the RVTS (Reviewer 1 / Comment 3). I would respectfully suggest that the paper could well include a reference to RVTS as an example of a successful longstanding program based on telesupervision in remote areas, but am happy to leave this decision to the discretion of the editors / authors.

Reviewer #2: Again thanks for undertaking this important work regarding telemonitoring and for taking the comments into consideration. The authors made the minimum revisions required, it could have been more argued.

7. PLOS authors have the option to publish the peer review history of their article (what does this mean?). If published, this will include your full peer review and any attached files.

Reviewer #1: **Yes: **Prof Tarun Sen Gupta

Reviewer #2: No

---

## [Editor Report · Acceptance letter]

3 Jul 2023

PONE-D-22-33318R1 

Characteristics of perceived effective telesupervision practices: A case study of supervisees and supervisors 

Dear Dr. Martin:

I'm pleased to inform you that your manuscript has been deemed suitable for publication in PLOS ONE. Congratulations! Your manuscript is now with our production department. 

Kind regards, 

on behalf of

Professor Federica Canzan 

Academic Editor

PLOS ONE